# Transplantation of human iPSC-derived muscle stem cells in the diaphragm of Duchenne muscular dystrophy model mice

Yasutomo Miura[1,2], Mase Sato[1], Toshie Kuwahara[3], Tomoki Ebata[2], Yasuhiko Tabata[3], Hidetoshi Sakurai[1]*

1 Department of Clinical Application, Center for iPS Cell Research and Application (CiRA), Kyoto University, Sakyo-ku, Kyoto, Japan, 2 Divison of Surgical Oncology, Department of Surgery, Nagoya University Graduate School of Medicine, Showa-ku, Nagoya, Japan, 3 Laboratory of Biomaterials, Department of Regeneration Science and Engineering, Institute for Frontier Life and Medical Science, Kyoto University, Sakyo-ku, Kyoto, Japan

* hsakurai@cira.kyoto-u.ac.jp

**Data Availability Statement:** All relevant data are within the paper and its Supporting Information files.

## Abstract

Duchenne muscular dystrophy (DMD) is an intractable genetic muscular disorder characterized by the loss of DYSTROPHIN. The restoration of DYSTROPHIN is expected to be a curative therapy for DMD. Because muscle stem cells (MuSCs) can regenerate damaged myofibers with full-length DYSTROPHIN *in vivo*, their transplantation is being explored as such a therapy. As for the transplanted cells, primary satellite cells have been considered, but donor shortage limits their clinical application. We previously developed a protocol that differentiates induced pluripotent stem cells (iPSCs) to MuSCs (iMuSCs). To ameliorate the respiratory function of DMD patients, cell transplantation to the diaphragm is necessary but difficult, because the diaphragm is thin and rapidly moves. In the present study, we explored the transplantation of iMuSCs into the diaphragm. First, we show direct cell injection into the diaphragm of mouse was feasible. Then, to enhance the engraftment of the transplanted cells in a rapidly moving diaphragm, we mixed polymer solutions of hyaluronic acid, alginate and gelatin to the cell suspension, finding a solution of 20% dissolved hyaluronic acid and 80% dissolved gelatin improved the engraftment. Thus, we established a method for cell transplantation into mouse diaphragm and show that an injectable hyaluronic acid-gelatin solution enables the engraftment of iMuSCs in the diaphragm.

## Introduction

Duchenne muscular dystrophy (DMD) is an X-linked genetic disease due to a mutation of *DMD* gene, which encodes DYSTROPHIN. DYSTROPHIN is part of the protein complex that connects the actin cytoskeleton to the extracellular matrix and contributes to muscle membrane stability [1]. DMD patients are usually diagnosed at 3–5 years old and lose walking ability at about 12 years old. In addition, patients will need respiratory support and suffer from cardiomyopathy in their early teens [2]. The main causes of death are heart failure, respiratory

**Funding:** This work was supported by a grant from the Core Center for iPS Cell Research (JP21bm0104001), Research Center Network for Realization of Regenerative Medicine from the Japan Agency for Medical Research and Development (AMED) to H.S. The funders had no role in study design, data collection and analysis, decision to publish, or preparation of the manuscript.

**Competing interests:** The authors have declared that no competing interests exist.

failure, and respiratory infection [3]. Currently, there is no curative treatment, and palliative therapy, such as corticosteroids, is used [4].

The restoration of DYSTROPHIN is a promising therapy for DMD, and several therapeutic approaches have been investigated, including exon-skipping oligonucleotides [5, 6], gene therapy [7] and cell transplantation [8]. While the other strategies aim to restore truncated versions of DYSTROPHIN, cell transplantation potentially regenerates myofibers with full-length DYSTROPHIN. Previous research on cell transplantation showed that the intramuscular injection of myoblasts and satellite cells regenerates normal myofibers in mice [9, 10] However, there is a shortage of muscle stem cell (MuSC) donors, and the cells are difficult to expand while maintaining their stemness and engraftment ability prior to the injection [11, 12]. As an alternative, the differentiation of induced pluripotent stem cells (iPSCs) into engraftable iPSC-derived MuSCs (iMuSCs) has been explored [13].

To ameliorate respiratory function in DMD, transplantation to the diaphragm, which is the main respiratory muscle, is necessary. MuSCs cannot pass through the vessel wall in intra-arterial transplantation [14], which makes direct cell injection into the muscle necessary. However, direct cell injection to the diaphragm is difficult due to the muscle's thin dimension. Cell-sheet attachment is considered an effective method for cell delivery to the diaphragm [15], but the sheets fail to couple electromechanically with the host muscle [16].

In the present study, we investigated cell transplantation into the diaphragm by direct injection in DMD model mice. To enhance the engraftment efficiency, we injected the cells with a mixed polymer solution of hyaluronic acid and gelatin or alginate and gelatin to promote cell engraftment and proliferation *in vivo* [17, 18]. Our results provide a strategy for direct cell transplantation into the diaphragm of mice.

## Materials and methods

### Study approval

All animal experiments were performed in accordance with the guidelines of the animal experiment committee and recombinant DNA experiment committee at Kyoto University. The study design was reviewed and accepted by the CiRA Animal Experiment Committee prior to conducting any experiments (No.20-232).

### Animals

The NOG-mdx mouse strain (NOD.Cg-*Prkdc^scid^ Dmd^mdx^ il2rg^tm1Sug^*/Jic), which is an immunodeficient DMD model mice, was provided by CIEA (Kanagawa, Japan). C57BL/6NCrSlc (B6) mice were purchased from Simizu Laboratory Supplies Co., Ltd. (Kyoto, Japan). Transplantation experiments were performed on NOG-mdx mice 8–12 weeks old. C57BL/6-Tg (CAG-EGFP) [19] mice were purchased from Japan SLC (Sizuoka, Japan). Primary satellite cells were harvested from CAG-EGFP mice 6–7 weeks old and used as donor cells.

### Cells

**Primary satellite cells.** A previous study reported that the monoclonal antibody SM/C 2.6, which is a specific mouse cell-surface marker of quiescent satellite cells and was kindly provided by Dr. So-ichiro Fukada (Osaka University), can be used to purify satellite cells from mouse skeletal muscle by flow cytometry (FCM) [20]. Muscle fibers were harvested from the limbs of CAG-EGFP mice 6–7 weeks old and dissociated with Collagenase type II (CLASSII, Worthington). SM/C 2.6 positive and CD31, CD45, Sca-1 triple negative cells were sorted using an Aria II (BD Biosciences) based on a previous report [20]. Isolated satellite cells from

CAG-EGFP mice were detected as the GFP positive population by FCM (S1 Fig). A list of the antibodies used is provided in S1 Table.

**Hu5/KD3.** Hu5/KD3 is an immortalized myoblast cell clone [21] and was kindly provided by Dr. Naohiro Hashimoto (National Center for Geriatrics and Gerontology). To visualize the cells by fluorescence microscopy, a GFP-transgenic Hu5/KD3 line was established by a GFP-expressing *piggyBac* vector, PB-EF1α-EiP [22]. Hu5/KD3 and GFP-transgenic Hu5/KD3 were maintained in DMEM (Invitrogen) media supplemented with 20% fetal bovine serum (FBS, Biosera), 2% Ultroser G (Pall Corp.), 1% L-glutamine (Invitrogen), and 0.5% Penicillin Streptomycin Mixed Solution (nacalai tesque) on a collagen-coated dish.

## Human iPSC culture and differentiation

The human iPSC lines Ff-WJ14s01 and 414C2 were used for the experiments. Ff-WJ14s01 (an HLA homozygous iPSC line with the most frequent haplotype in Japan, abbreviated as S01 in this manuscript) was established from cord blood cells by using episomal vectors, while 414C2 was established from purchased human dermal fibroblast as previously described [23]. S01 was generated under written consent with approval by the Kyoto University Graduate School and Faculty of Medicine Ethics Committee (approval numbers #E1762, #G567and #Rinsho71). Both S01 and 414C2 were labeled with a Venus reporter for *PAX7*, and then they were differentiated into iMuSCs as previously described [13, 24]. Briefly, the iPSCs were seeded in a Matrigel-coated 6-well plate and cultured with StemFit (AK02N, Ajinomoto) medium ($1x10^4$ cells/well). At day 3, the medium was changed to CDMi medium supplemented with CHIR99021 (CHIR, Axon MedChem, Tocris) and SB431542 (SB, Sigma). CDMi medium is composed of IMDM (Invitrogen) and F12 (1X) Nutrient Mixture (Ham) (Invitrogen) at the ratio 1:1 supplemented with 1% BSA (Sigma), 0.5% Penicillin Streptomycin Mixed Solution, 1% CD Lipid Concentrate (Invitrogen), 1% Insulin-Transferrin Selenium (Invitrogen) and 450 μM 1-Thioglycerol (Sigma). Seven days later, the cells were passed to a Matrigel-coated dish with CDMi medium supplemented with SB and CHIR ($1x10^6$ cells/6-cm dish). Seven days later, the cells were passaged to a Matrigel-coated 6-well plate with CDMi medium ($8x10^5$ cells/well). Three days later, the medium was switched to SFO3 medium (Sanko-Junyaku) supplemented with IGF-1, bFGF, and HGF. At day 38 of the differentiation, the medium was switched to DMEM (Invitrogen) supplemented with 0.5% Penicillin Streptomycin Mixed Solution, 2 mM L-glutamine (nacalai tesque), 0.1 mM 2-ME, 2% Horse Serum (HS, Sigma), 5 μM SB and 10 ng/mL IGF-1. This medium was replaced with fresh medium of the same composition 3 times per week until day 84 of the differentiation. Afterwards, mature myotubes were observed, and *PAX7*-VENUS positive iMuSCs were sorted by FCM.

## Flow cytometry sorting of muscle stem cells

After day 84 of the iMuSC induction, the cells were dissociated and sorted for the Venus reporter by FCM. The cells were dissociated with Collagenase G (500 μg/mL) (meiji), Collagenase H (100 μg/mL) (meiji) and DISPASE II (0.3 mg/mL) (GODO SHUSEI) for 60 minutes, followed by 5 minutes with Accutase (Innovative Cell Technologies, Inc.) at 37˚C and then filtered with a 40-μm mesh. To remove non-viable cells from the cell suspension, the suspension was layered on an OptiPrep (Merck) with DMEM containing 10% serum and centrifuged at 800x*g* for 20 minutes. The density of OptiPrep and DMEM containing 10% serum was higher than that of the viable cells, and after the centrifugation, non-viable cells were pelleted, macromolecular material or enzymes remained in the supernatant, and viable cells were recognized as an interfacial band. Thus, viable cells were harvested at the interfacial band and diluted with DMEM containing 10% serum, followed by centrifugation at 400x*g* for 10 minutes. Finally, the

cells were resuspended with HBSS buffer containing 1% bovine serum albumin (Sigma) and Propidium Iodide (0.5 µg/mL) (Sigma). Venus-Pax7 positive cells were sorted using the Aria II.

## Preparation of injectable mixed polymer solution

Gelatin (Nitta Gelatin Inc.), hyaluronic acid (Denka) and alginate (KIMICA Inc.) were dissolved in injectable media, 1% w/v at 37˚C. Polymer solutions were prepared by mixing alginate with gelatin (A/G solution) or hyaluronic acid with gelatin (H/G solution) and added to the cell suspension. The volumetric ratio of the injectable solution (mixed polymer solution and cell suspension) was 4:1.

## Measurement velocity of polymer solution

Each A/G and H/G solution was dropped onto a tilted glass slide, and the distance travelled by the solution in 10 seconds was measured. This measurement for each polymer solution was performed 3 times repeatedly.

## Transplantation to the diaphragm

All mice were anesthetized using 2% Isoflurane. Laparotomy was performed, and cell suspension with or without mixed polymer solutions was injected with a syringe and 33G needle (ITO CORPORATION) at the edge of the diaphragm by stereomicroscopy via the abdomen. Freshly sorted cells were resuspended, and $1 \times 10^4$ cells per injection site (primary satellite cells) or $1 \times 10^5$ cells per injection site (GFP-transgenic Hu5/KD3 and iMuSCs) were injected. The injection volume per site was 10 µL. Injection into the diaphragm was recognized by swelling of the injection site.

## Transplantation to the tibialis anterior

All mice were anesthetized using isoflurane. After exposure of the tibialis anterior (TA), cryoinjury [24] was performed 3 times by direct contact for 12 seconds each time, followed by the transplantation of $1 \times 10^6$ GFP-transgenic Hu5/KD3 cells per TA.

## Immunocytochemistry

Cultured cells were fixed with 2% Paraformaldehyde (PFA, nacalai tesque) in Phosphate buffer solution (PBS, nacalai tesque) for 10 min on ice, washed 2 times with PBS for 10 minutes and blocked with Blocking One (nacalai tesque) for 1 hour at 4˚C on a shaker. Then, the samples were incubated with primary antibodies diluted in 10% Blocking One/PBS containing 0.2% Triton-X (nacalai tesque) (PBS-T) overnight at 4˚C. The following day, the samples were washed 3 times for 10 minutes with 0.2% PBS-T and incubated with the corresponding secondary antibodies and DAPI (1:5000) at room temperature on the shaker. Finally, the samples were observed under a BZ-X700 microscope (KEYENCE).

A list of the antibodies used is provided in S1 Table.

## Immunohistochemistry

The TA and diaphragm were harvested and cryopreserved. Cryosections (10–12 µm) were fixed with 4% PFA for 20 min, washed 2 times with PBS for 5 minutes, blocked with Blocking One (nacalai tesque) for 1 hour at room temperature, and incubated with primary antibodies diluted in Can get Signal B solution (TOYOBO) overnight at 4˚C. The samples were washed 3 times for 10 minutes with 0.2% PBS-T and incubated with the corresponding secondary

antibodies and DAPI (1:5000) for 1 hour at room temperature. Then the slides were washed 1 time for 5 minutes with PBS-T and 2 times for 5 minutes with PBS. After that, the samples were mounted in Aqua-Poly-Mount mounting medium (Polyscience). Sections were observed under an LSM710 (Carl Zeiss) or FV3000 (Olympus) confocal microscope. To detect human nuclei and myofibers by optical microscopy, the cryosections were stained with 3,3'-Diamino-benzidine (DAB) and eosin. The cryosections were also stained with anti-h-Nuclei antibody as the primary antibody and anti-mouse IgG1 HRP antibody, followed by 3 washes with PBS-T for 10 minutes each time. Then, DAB staining was performed using the Peroxidase Stain DAB Kit (nacalai tesque) according to the manufacturer's protocol. After that, the samples were washed with water for 20 minutes, stained with 0.5% Eosin Y-solution (Merck) for 6 minutes, and washed with water again for 6 minutes. The samples were mounted in MOUNT-QUICK (DAIDO SANGYO) and observed under a BX51 microscope (Olympus).

A list of the antibodies used is provided in S1 Table.

## Observation of GFP signals

Mice were sacrificed, and the diaphragm was harvested and immediately observed under a fluorescence stereomicroscope (Leica M205FA). After the observation, the sample was fixed with 4% PFA for 30 minutes at 4˚C and 30% sucrose overnight at 4˚C, followed by cryopreservation.

## Statistical analysis

All statistical analyses were performed using GraphPad Prism version 9.2.0 for Mac OS (GraphPad Software). Data are expressed as the mean ± SEM. Differences between two groups were analyzed using the two-tailed Student's t test, and differences between three or more groups were analyzed by ANOVA (analysis of variance) with Tukey's range test for multiple comparisons. Significant differences were considered as $p < 0.05$.

## Engraftment efficiency analysis

The TA and diaphragm were sectioned every 10–12 μm and stained. Myofibers derived from engrafted cells were recognized as h-SPECTRIN positive fibers. To quantify the number of h-SPECTRIN positive fibers, TA samples 2 weeks after the transplantation and diaphragm samples 4 weeks after were analyzed using an SM710 (Carl Zeiss) or FV3000 (Olympus) confocal microscope. The number of detected h-SPECTRIN-positive fibers in one section was recorded as the number of engrafted myofibers.

## Validation of cell proliferation *in vitro* after passing through the transplantation needle

The condition of passaging was divided into three groups: passing through the 33G needle without polymer solution and then cultured, passing through the needle with polymer solution and then cultured, and cultured without passing through the needle. Three types of cells, primary satellite cells, Hu5/KD3 and iMuSCs, were seeded into 3 wells each. The number of cells was counted by trypan blue staining before cell confluency, and the cell proliferation was compared within the three groups.

## Results

### Primary satellite cells regenerate dystrophic diaphragm in NOG-mdx mice

To investigate if cell transplantation into the diaphragm is feasible, we used mouse primary satellite cells. After isolating primary satellite cells from the limb muscles of CAG-EGFP mice

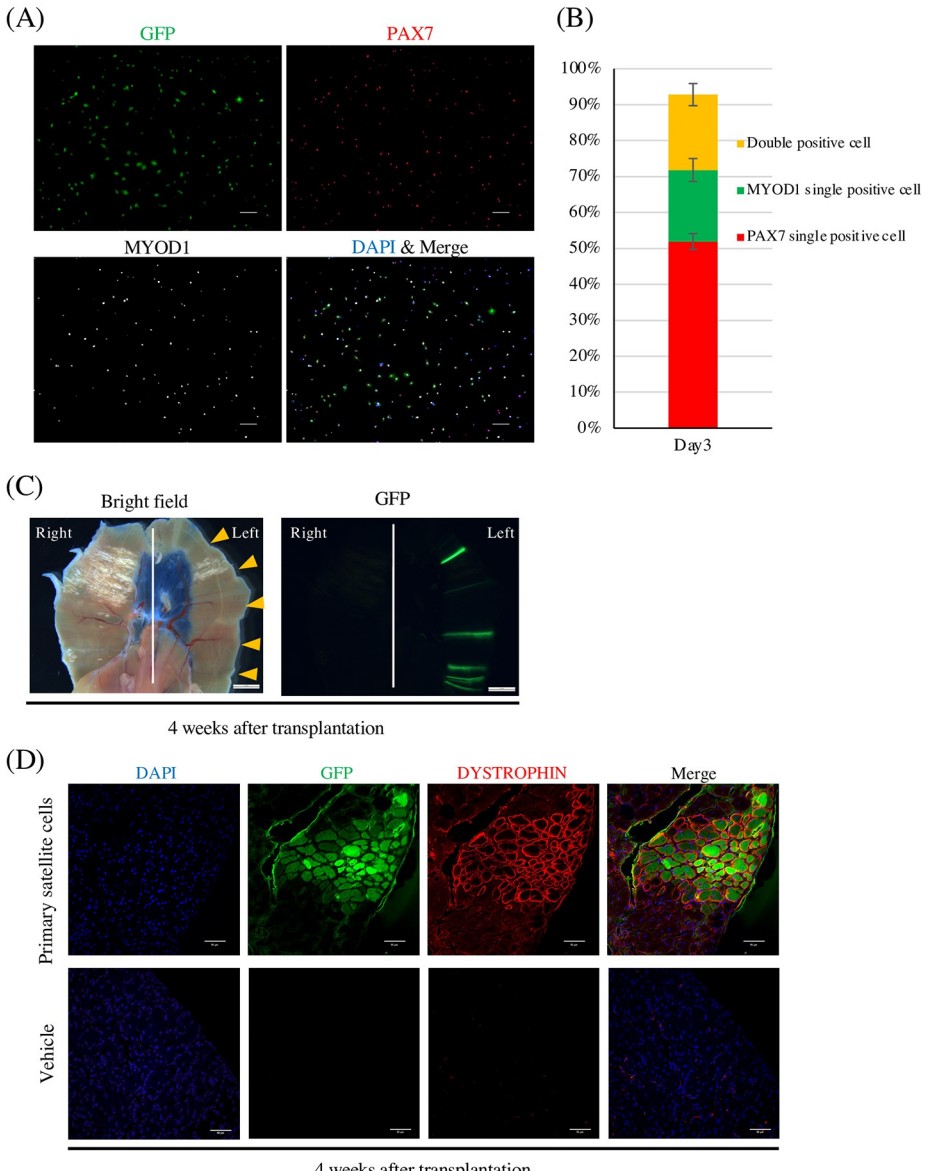

**Fig 1. Primary satellite cells engrafted in the diaphragm.** (A) Immunocytochemistry staining of isolated primary satellite cells after 3 days of culture. GFP (green), DAPI (blue), PAX7 (red), MYOD1 (white). Scale bars, 100 μm. (B) Quantification of PAX7+MYOD–, PAX7+MYOD+, and PAX7–MYOD+ cell populations after 3 days of culturing primary satellite cells. Data show the mean ± SD. n = 3. (C) Representative images of the diaphragm by fluorescence stereomicroscopy 4 weeks after the transplantation. The cells were injected to the left side of the diaphragm, and vehicle was injected to the right side. Scale bars, 2 mm. Arrowheads show injection sites. (D) Representative immunohistochemistry staining of a cryosection. Sections were stained with anti-DYSTROPHIN antibody (red) and anti-GFP antibody (green). Nuclei were stained with DAPI (blue). Scale bars, 50 μm.

by FCM, the purity of the isolated cells was validated by myogenic marker expression after 3 days of culture (Fig 1A). Isolated cells were GFP positive (S1 Fig). More than 90% of cells were PAX7 and/or MYOD1 positive (Fig 1B), indicating good purity for the transplantation experiment. The cells were then transplanted into NOG-mdx mice by direct injection through the edge of the diaphragm. GFP signals were detected in the transplanted area (Fig 1C), and DYSTROPHIN positive fibers expressing GFP derived from the transplanted cells were detected

(Fig 1D). In total, these results demonstrated that MuSC transplantation into the diaphragm of mice is feasible.

## Transplantation of human immortalized myoblasts into the diaphragm shows limited engraftment

To confirm if human myogenic cells can be transplanted into the diaphragm, Hu5/KD3 [21], an immortalized human myoblast line, was tested. Hu5/KD3 has the potential to engraft into the skeletal muscle of immunodeficient mice to induce muscle regeneration *in vivo* [21]. To visualize the engrafted area as a whole mount sample, GFP-transgenic Hu5/KD3 was generated and transplanted into the diaphragm of NOG-mdx mice. GFP was expressed homogenously, and the cells were MYOD1 positive but PAX7 negative (Fig 2A). Although GFP was expressed in whole mount samples (Fig 2B upper panel) and DYSTROPHIN positive fibers were detected 4 weeks after the transplantation (Fig 2B lower panel), the magnitudes were less than with the transplantation of mouse primary cells (Fig 2C), suggesting Hu5/KD3 cells need further modification to enhance their engraftment potential in the diaphragm of NOG/mdx mice.

## Cell suspension with mixed polymer solution improves the engraftment efficiency of Hu5/KD3

During the cell transplantation, we noticed that the injected cell suspension flowed out of the diaphragm due to rapid respiratory movement. We hypothesized that this effect could contribute to the low engraftment efficiency. Previous studies on cardiomyocyte transplantation have reported that gelatin enhances the engraftment of transplanted cells [17, 25] and that a mixed polymer solution composed of alginate and gelatin with ferric irons improves cell survival and proliferation *in vitro* and *in vivo* [26]. In addition to gelatin and alginate, hyaluronic acid is a viscous material that has been used in clinical practice [27].

We hypothesized that the viscosity of the mixed polymer solution may affect cell migration. We therefore evaluated the viscosity of mixed polymer solutions of different hyaluronic acid/gelatin and alginate/gelatin composition on tilted glass (Fig 3A). To identify the most suitable mixed polymer solution condition for myogenic cell transplantation, Hu5/KD3 with mixed polymer solution was injected into the tibialis anterior (TA) muscle of NOG-mdx mice. Two weeks after the transplantation, h-SPECTRIN positive fibers were counted to assess the engrafted cells because the SPECTRIN signal was stronger than the DYSTROPHIN signal (Fig 3B arrowheads). H2G8 (20% dissolved hyaluronic acid and 80% dissolved gelatin) and A2G8 (20% dissolved alginate and 80% dissolved gelatin) improved the engraftment efficiency significantly (Fig 3C and 3D). Compared to medium condition, A10 and A5G5 reduced the engraftment efficiency significantly. Moreover, a large gap was observed between host myofibers with the A10 (100% dissolved alginate) transplantation, suggesting alginate prevented migration of the engrafted cells because it is not biodegradable, unlike hyaluronic acid and gelatin (Fig 3E).

## Mixed hyaluronic acid-gelatin solution reduces the frequency of iMuSC engraftment failure in the diaphragm of NOG-mdx mice

To investigate if mixed polymer solutions can improve the engraftment efficiency of cell transplantation into the diaphragm, GFP-transgenic Hu5/KD3 with A2G8, H2G8 or no polymer solution (Medium) was injected. However, we found no significant difference between the three groups (Fig 4A and 4B). Next, the effect of the mixed polymer solutions on the iMuSC transplantation was assessed. We isolated iMuSCs by FCM after differentiating iPSCs with a

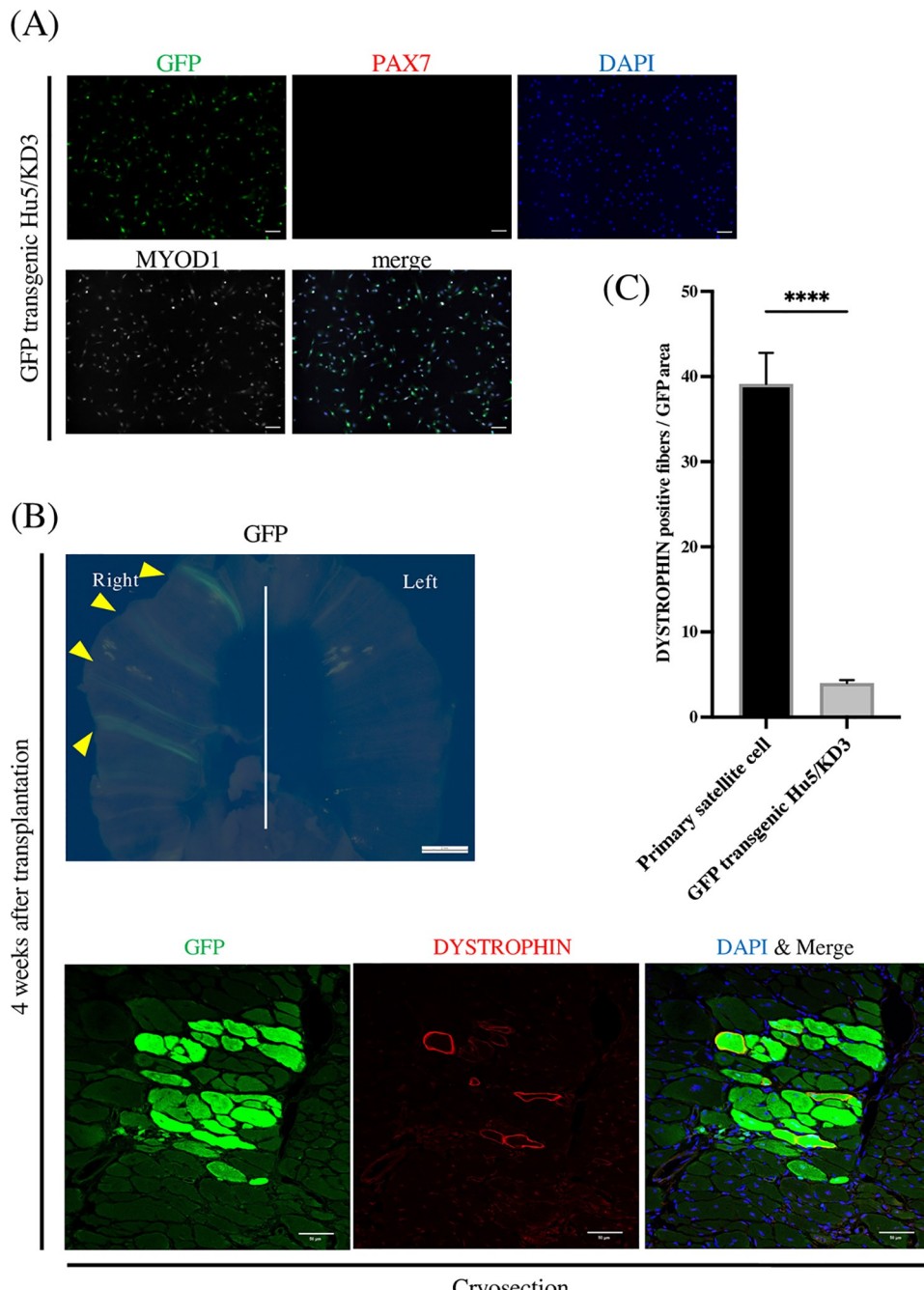

**Fig 2. Human immortalized myoblasts in the diaphragm show limited engraftment.** (A) Immunocytochemistry staining of GFP-transgenic Hu5/KD3 in culture. Cells were stained with anti-PAX7 antibody (red), anti-MYOD1 antibody (white), and DAPI (blue). Scale bars, 100 μm. (B) A representative fluorescence stereomicroscopy image of the diaphragm 4 weeks after the transplantation of GFP-transgenic Hu5/KD3 (upper panel). Scale bar, 2 mm. Arrowheads show GFP. Immunohistochemistry staining of the image (lower panel). The image was stained with anti-DYSTROPHIN antibody (red) and anti-GFP antibody (green). Nuclei were stained with DAPI (blue). Scale bars, 50 μm. (C) Quantification of DYSTROPHIN positive fibers in the transplanted areas. Data show the mean ± SEM (two-tailed Student's t-test; ****p<0.0001) and are from three independent experiments. Three mice were used in one experiment. There were in total 87 primary satellite cells and 78 GFP transgenic Hu5/KD3 cells in the respective GFP areas.

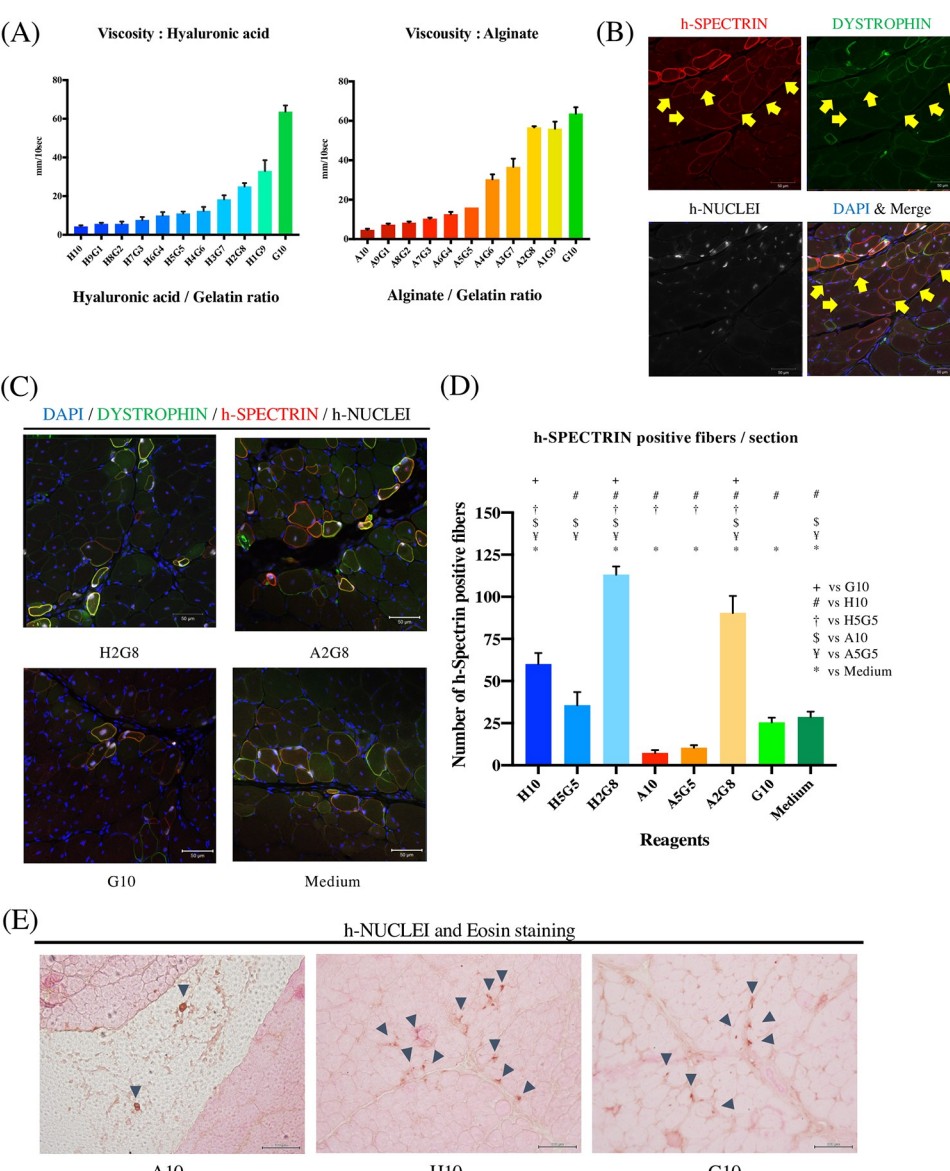

**Fig 3. Injectable mixed polymer solution improves engraftment efficiency in the TA.** (A) Velocities of mixed polymer solutions along a tilted glass. Data shown are from three independent experiments. (B) Immunohistochemistry staining 2 weeks after Hu5/KD3 transplantation into the TA. Sections were stained with anti-DYSTROPHIN antibody (green), anti-h-SPECTRIN antibody (red) and anti-h-NUCLEI antibody (white). Nuclei were stained with DAPI (blue). Scale bars, 50 μm. Yellow arrowheads show the DYSTROPHIN signal was weaker than the signal in myofibers. (C) Immunohistochemistry staining 2 weeks after Hu5/KD3 transplantation into the TA. Sections were stained with anti-DYSTROPHIN antibody (green), anti-h-SPECTRIN antibody (red) and anti-h-NUCLEI antibody (white). Nuclei were stained with DAPI (blue). Scale bars, 50 μm. (D) Quantification of h-SPECTRIN positive fibers derived from the engrafted cells. Data show the mean ± SEM (ANOVA with Tukey's multiple comparison test). Each mark above the bars indicates a statistically significant difference. Data shown are from three independent experiments. The number of sections in each analysis is 44 or 45. (E) Immunohistochemistry and eosin staining. Engrafted cells were stained with h-NUCLEI (blown). Arrowheads show engrafted cells. Scale bars, 100 μm. For the mixed polymer solution names, H = hyaluronic acid, A = alginate, G = gelatin, and numbers indicate the ratios of the material in the mixed polymer solution.

knock-in Venus reporter for *PAX7* gene (S2 Fig) and transplanted them into the diaphragm of NOG-mdx mice with H2G8, A2G8 or culture medium [13, 24]. Four weeks after the transplantation, the iMuSC engraftment was analyzed by the number of h-SPECTRIN positive

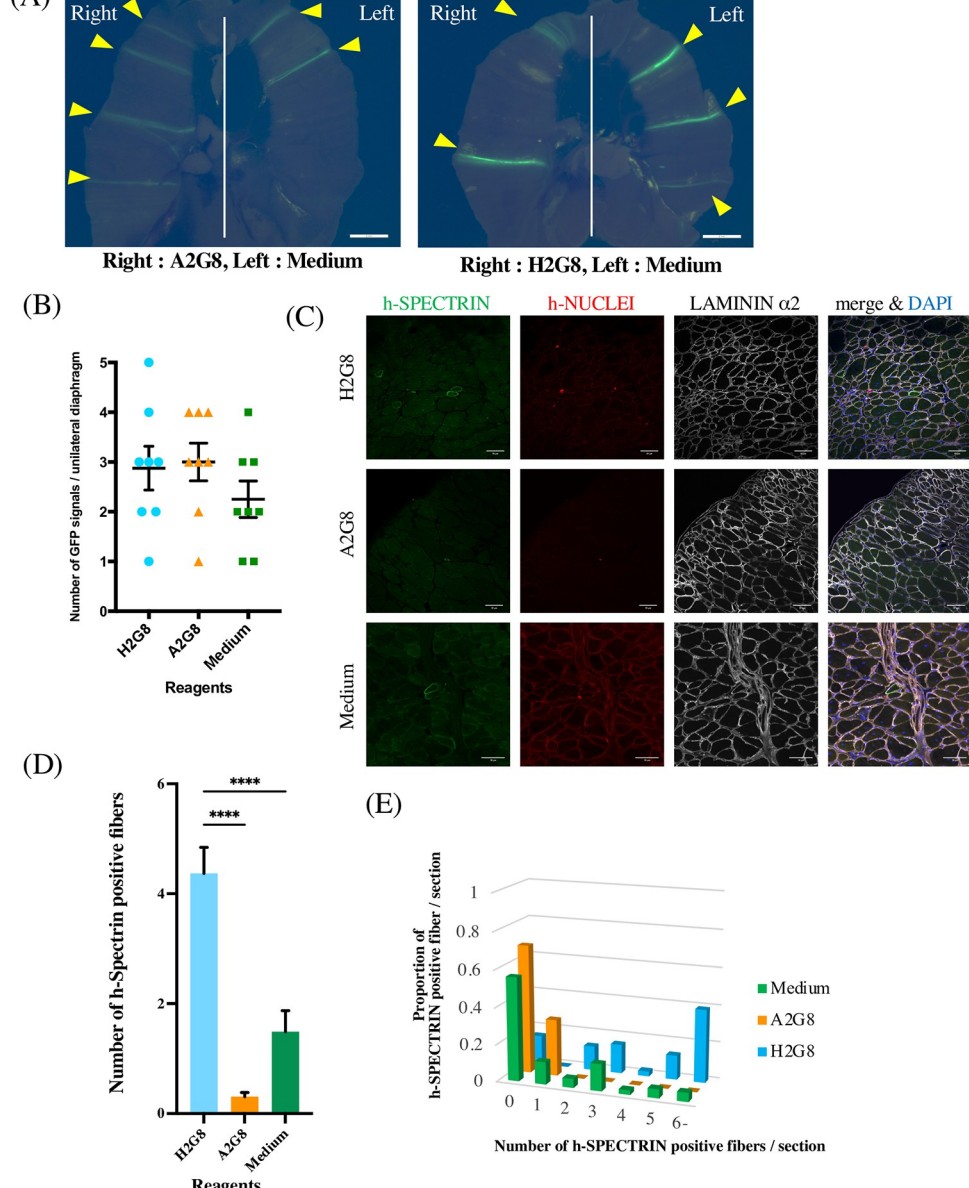

**Fig 4. H2G8 reduces the frequency of iMuSC engraftment failure in the diaphragm of NOG-mdx mice.** (A) A fluorescence stereomicroscopy image of the diaphragm 2 weeks after the transplantation of GFP-transgenic Hu5/KD3. Scale bars, 2 mm. Arrowheads show GFP. Data shown are from five independent experiments. Each of ten unilateral diaphragms were analyzed in the three groups (Medium, A2G8, H2G8). (B) Quantification of GFP signals in the unilateral diaphragm. Horizontal and vertical lines show means ± SEM (ANOVA with Tukey's multiple comparison test). (C) Immunohistochemistry staining 4 weeks after the iMuSC transplantation. Sections were stained with anti-h-SPECTRIN antibody (green), anti-h-NUCLEI antibody (red), and anti-LAMININ α2 antibody (white). Nuclei were stained with DAPI (blue). Scale bars, 50 μm. (D) Quantification of h-SPECTRIN positive fibers 4 weeks after the iMuSC transplantation. Data show the mean ± SEM (ANOVA with Tukey's multiple comparison test). ****p<0.0001. The number of sections is 38 (H2G8), 36 (A2G8) and 41 (Medium). (E) A histogram of h-SPECTRIN positive fibers per cryosection. The X-axis represents the number of h-SPECTRIN positive fibers in a single section, and the Y-axis represents the number of h-SPECTRIN positive fibers in a single section over all sections.

myofibers (Fig 4C). While the inclusion of H2G8 significantly increased the number of h-SPECTRIN positive fibers, the iMuSC engraftment was still very low (Fig 4D). However, when we assessed the proportion of h-SPECTRIN positive fibers per section, we found H2G8

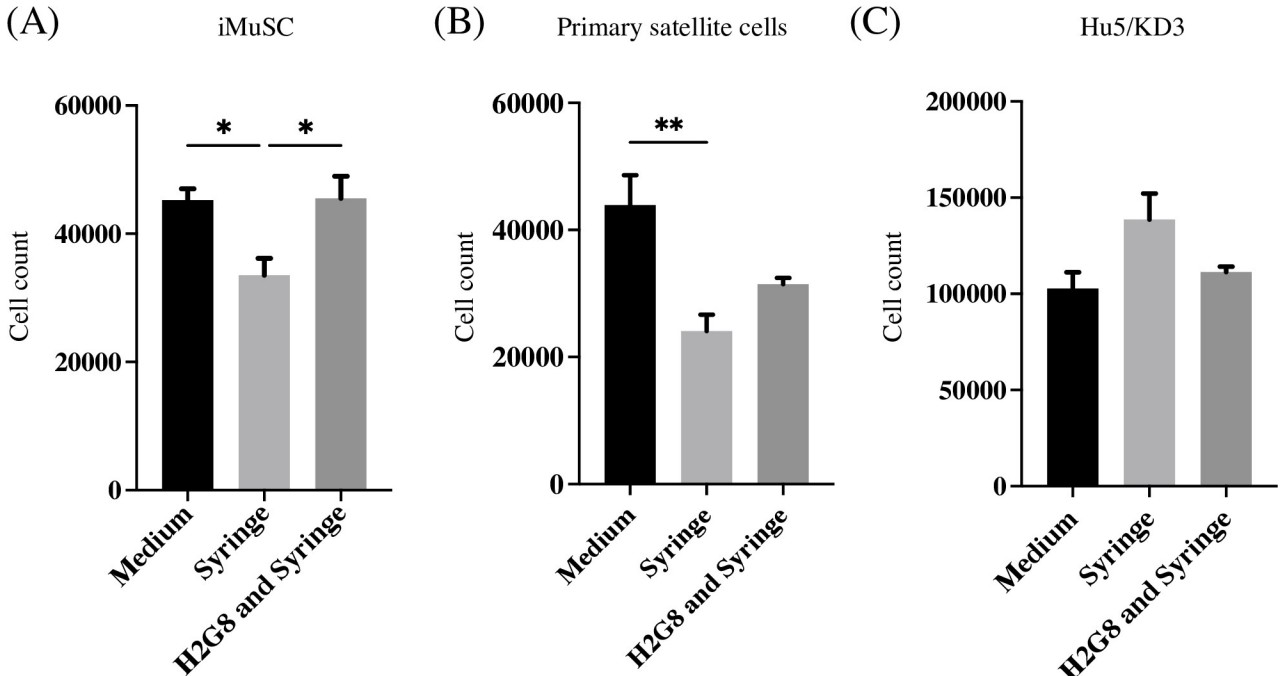

**Fig 5. H2G8 increases the proliferation of iMuSCs.** (A) Number of iMuSCs 3 days after passage. The seeding cell number was $1.5 \times 10^4$ cells/well. 'Syringe' describes cells passaged after passing through the needle; 'H2G8 and Syringe' describes cells passaged after passing through the needle with H2G8; and 'Medium' describes cells passaged without passing through the needle. (B) Number of mouse primary satellite cells 3 days after passage. The seeding cell number was $2.1 \times 10^4$ cells/well. (C) Number of Hu5/KD3 cells 3 days after passage. The seeding cell number was $2.1 \times 10^4$ cells/well. Data show the mean ± SEM (ANOVA with Tukey's multiple comparison test). *$p<0.05$, **$p<0.1$; n = 3.

reduced the frequency of engraftment failure, as the proportion of no h-SPECTRIN positive fibers was 0.15 with H2G8 but 0.69 and 0.56, respectively, with A2G8 and Medium (Fig 4E).

## Mixed hyaluronic acid-gelatin solution improves the proliferation of iMuSCs

It is reported that mechanical stress during flow through a transplantation needle affects cell viability and proliferation [28]. Since we used a 33G needle, cell proliferation after passing through the needle was analyzed with or without H2G8. Passing through the needle reduced the proliferation of iMuSCs in culture without H2G8. However, H2G8 improved the proliferation to that without the needle flow (Fig 5A). Mouse primary satellite cells also showed reduced proliferation after passing through the needle but did not benefit significantly from H2G8 (Fig 5B). On the other hand, the proliferation of Hu5/KD3 was not affected by passing through the needle, which might be due to the immortality of this line (Fig 5C).

## Discussion

In this study, we demonstrated iMuSC transplantation into the diaphragm of DMD model mice. First, we showed that transplantation by direct injection was feasible. Then, we found that a polymer mixture composed of gelatin and hyaluronic acid improved the engraftment efficiency.

Our results using mouse primary satellite cells demonstrated that stereomicroscopy with a tiny needle (33G) for the transplantation resulted in MuSCs regenerating skeletal muscle with intact DYSTROPHIN in the diaphragm. To ameliorate the respiratory function of DMD

patients, the transplantation of MuSCs to the diaphragm is necessary, and iMuSCs will solve any donor shortages. However, current iMuSC differentiation methods have poor efficiency and need a long time. Therefore, we first investigated our transplantation protocol using immortalized human myoblasts, Hu5/KD3, which, like iMuSCs, are much easier to proliferate than primary MuSCs and can engraft into the skeletal muscle of immunodeficient mice to regenerate skeletal muscle. Although GFP-transgenic Hu5/KD3 engrafted into the diaphragm, the number of regenerated myofibers derived from the engrafted cells was less than that of primary satellite cells. It is well known that freshly isolated primary satellite cells have vigorous regeneration potential, but cultured primary satellite cells show lower regeneration potential [12]. Since Hu5/KD3 cells are PAX7-negative, MYOD-positive myoblasts, the difference of the regeneration potential between freshly isolated satellite cells and myoblasts might explain their low engraftment efficiency. Another possibility is differences in the extracellular matrix between *in vivo* satellite cells and *in vitro* cultured Hu5/KD3, which may affect the proliferation [29, 30].

In addition, the movement of the diaphragm might affect the engraftment efficiency. The diaphragm contracts very quickly, especially in mice, who have a respiration rate of 150–200 beats per minute. Contraction causes the injection medium to flow out of the injection hole. We hypothesized that cell retention at the injection site is important for the engraftment efficiency and viscous materials, such as polymer solutions and hydrogels, may enhance the cell retention. Gelatin, alginate and hyaluronic acid are viscous materials used in clinical practice. A previous report showed that transplanting murine pre-osteoblasts with a mixed polymer solution of gelatin and alginate that physically interacts with ferric ions to improve cell retention in the transplanted area [26]. Another study reported that hyaluronic acid increases the viability of neural stem cells *in vivo* [31]. We tested the engraftment efficiency with injectable polymer solution by transplanting Hu5/KD3 into the TA and found two types, H2G8 (20% dissolved hyaluronic acid and 80% dissolved gelatin) and A2G8 (20% dissolved alginate and 80% dissolved gelatin), improved the engraftment efficiency significantly. In contrast, the number of h-SPECTRIN positive fibers seen when including A10 (100% dissolved alginate) or A5G5 (50% dissolved alginate and 50% dissolved gelatin) was much less, and in the case of A10 the engrafted cells were detected in gaps between the myofibers, which indicated that cell migration was prevented because alginate is not biodegradable *in vivo*.

Cell transplantation into the diaphragm of mice has never been reported. Indeed, we found it difficult to transplant iMuSCs into the diaphragm, and more than half of our experiments resulted in no engraftment. In contrast, for the Hu5/KD3 transplantation, all experiments resulted in at least one engraftment site if using the same transplantation procedure for primary satellite cells. These observations suggested the cell viability of Hu5 and iMuSCs after the transplantation was different. Hu5/KD3, which is immortalized by the transduction of hTERT, CDK4R24C and cyclin D1, showed no change in cell proliferation after passing through the transplantation needle, whereas both primary satellite cells and iMuSCs showed reduced proliferation. Furthermore, because iMuSCs are not immortalized and not obtained *in vivo*, they may have more difficulty to engraft in the diaphragm after passing through the transplantation needle. H2G8 benefited the engraftment slightly, suggesting H2G8 might prevent cell damage during the injection. Therefore, the inclusion of H2G8 is one step in the optimization of iMuSC transplantation into the diaphragm.

Regardless, despite the inclusion of H2G8, the engraftment efficiency of iMuSCs was still much less than that of primary satellite cells. Moreover, in the case of the diaphragm, H2G8 only reduced the frequency of iMuSC engraftment failure. Further optimization of the cell transplantation, such as the addition of some factor, such as glutamine, Wint7a [15], glycine [32], or uPAR [33], and higher purification of the iMuSCs are warranted [34].

## Conclusion

In summary, we demonstrated that cell transplantation into the diaphragm is feasible by direct injection and that the inclusion of a mixed hyaluronic acid-gelatin solution reduces the frequency of failed iMuSC transplantations into the diaphragm. These findings will contribute to the study of cell transplantations via local injections.

## Supporting information

**S1 Fig. FCM analysis of isolated primary satellite cells from mice.** (A) FCM analysis of isolated cells from B6 mice (left panel) and GFP expression (right panel). (B) FCM analysis of isolated primary satellite cells from CAG-EGFP mice (left panel) and GFP expression (right panel).
(TIF)

**S2 Fig. FCM analysis after differentiating iPSCs with a knock-in Venus reporter for *PAX7* gene.**
(TIF)

**S1 Table. A list of antibodies used.**
(TIF)

## Acknowledgments

We would like to thank Akitsu Hotta for providing GFP-expressing *piggy-Bac* vector; So-ichiro Fukada for providing SM/C 2.6 antibody and the frozen stock of mouse primary satellite cells; Kanae Mitsunaga for her constant help with the FACS experiments; Syunsuke Kihara for his help with the microscopy analysis; Megumi Goto for technical assistance; and Peter Karagiannis for proofreading the manuscript.

## Author Contributions

**Conceptualization:** Hidetoshi Sakurai.

**Data curation:** Yasutomo Miura.

**Funding acquisition:** Hidetoshi Sakurai.

**Investigation:** Yasutomo Miura, Mase Sato.

**Project administration:** Hidetoshi Sakurai.

**Resources:** Toshie Kuwahara, Yasuhiko Tabata.

**Supervision:** Tomoki Ebata, Yasuhiko Tabata, Hidetoshi Sakurai.

**Validation:** Hidetoshi Sakurai.

**Writing – original draft:** Yasutomo Miura.

**Writing – review & editing:** Hidetoshi Sakurai.

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
