## [Decision Letter · Decision Letter 0]

29 Sep 2021

PONE-D-21-27260Injectable hyaluronic acid-gelatin solution enhances the engraftment efficiency of human iPS cell-derived muscle stem cells in the diaphragm of Duchenne muscular dystrophy model micePLOS ONE

Dear Dr. Sakurai

Thank you for submitting your manuscript to PLOS ONE. After careful consideration, we feel that it has merit but does not fully meet PLOS ONE’s publication criteria as it currently stands. Therefore, we invite you to submit a revised version of the manuscript that addresses the points raised during the review process.

I am returning your manuscript with 2 reviews. The reviewers came to different conclusions about the article, as you will see. After reading the reviews and looking at the manuscript, my decision is "Major Revision". In particular, we would expect a revised manuscript to address cell viability after going through needles and experimental details (including N values, controls used and statistical analysis). In addition, the conclusions are not fully supported by the data and should be revised, as well as the title.This is not to say, however, that we consider any other concerns raised by our referees to be any less important.

We look forward to receiving your revised manuscript.

Kind regards,

Julie Dumonceaux

Academic Editor

PLOS ONE

Journal Requirements:

 [This work was supported by a grant from the Core Center for iPS Cell Research (JP21bm0104001), Research Center Network for Realization of Regenerative Medicine from the Japan Agency for Medical Research and Development (AMED) to H.S.]

Reviewers' comments:

Reviewer's Responses to Questions

**Comments to the Author**

1. Is the manuscript technically sound, and do the data support the conclusions?

Reviewer #1: Partly

Reviewer #2: No

2. Has the statistical analysis been performed appropriately and rigorously? 

Reviewer #1: No

Reviewer #2: Yes

3. Have the authors made all data underlying the findings in their manuscript fully available?

Reviewer #1: No

Reviewer #2: No

4. Is the manuscript presented in an intelligible fashion and written in standard English?

Reviewer #1: Yes

Reviewer #2: Yes

5. Review Comments to the Author

Reviewer #1: This is an interesting manuscript from an excellent laboratory addressing and important bottleneck in muscle cell therapy, i.e. delivery of myogenic cells to the diaphragm. The manuscript reads well and and the figures are also well presented (just occasional spelling mistakes).

My main concerns (which are easily addressable) are:

1) If diaphragm movements affect cell engraftment, why is this not impacting on mouse MuSC engraftment as well?

2) Authors use a 33G needle for their injections. This is a very small gauge which could potentially lysate the cell suspension being delivered. As human cells are larger in size than mouse cells, this could potentially explain the difference in engraftment. So the authors should perform a simple experiment of cell viability to assess cell death or cell lysis before and after passing through such a small needle, to make sure cell size and cell concentration/density does not negatively impact on the final outcome.

3) Graphs with statistical analysis of transplants need to specify the data points / N values as otherwise is difficult to interpret those numbers.

4) The main finding of the study might be overstated in the title, as in actual fact the absolute differences in dystrophin positive fibers per diaphragm is minimal, as shown in figure 4 d-e. It also unclear how such small numbers might result in such a large statistical significance: please clarify, add N values and consider amending the title into something more factual/descriptive/cautious (e.g. Transplantation and engraftment dynamics of mouse and human myogenic cells in the diaphragm of a mouse model of Duchenne muscular dystrophy).

Reviewer #2: In this manuscript, Miura and colleagues report a technical strategy to enhance the engraftment efficiency of iPS cell-derived muscle stem cells in the diaphragm of a mouse model for Duchenne muscular dystrophy by using an injectable hyaluronic acid-gelatin solution. Although the topic is of interest, the manuscript is of poor quality, presenting many weaknesses and inconsistencies. In addition, this seems more like an initial draft of a manuscript than the final version (missing experimental details and importantly, barely describing results).

Specific comments:

1- Fig. 1a: It is impossible to see the presence of PAX7 positive cells as alluded in Fig. 1b.

2- There is no evidence that the satellite cells are expressing GFP after sorting. The authors are not using GFP to sort satellite cells, and do not show GFP after 3 days in culture.

3- For the transplantation involving satellite cells and myoblasts, the authors quantify the number of fibers per GFP area. Does it correspond to each injection point? The GFP signal looks greater in the image showing the whole diaphragm when compared to the actual number of fibers (autofluorescence?).

4- Fig. 1c-d: It is hard to tell where exactly cells were injected (panel c, left). Also, many GFP+ myofibers are dystrophin negative. How do the authors explain this discrepancy? How about Pax7 staining? Not clear what we should be looking at for this. In addition, this experiment is missing proper controls.

5- Page 14, line 221: The authors refer to unpublished results for the proof-of-concept that used immortalized myoblasts are able to contribute to muscle regeneration in vivo, but this information seems critical for the present manuscript since the authors barely see engraftment in the diaphragm (5 fibers is negligible). The authors barely describe these results.

6- Figure 2: Same issue as figure 1. Most fibers do not express DYSTROPHIN. The number of mice per experimental group is missing.

7- Experiments with hyaluronic acid-gelatin solution are extremely weak. The authors inject immortalized myoblast into the TA to identify this combination, then, when translating into the diaphragm, they use myoblast first and just quantify the GFP signal, they don t show the number of fibers. Why testing human immortalized myoblasts into the TA instead of the diaphragm? There is no evidence that the TA needs extra-help since it is not a moving muscle. This brings another point, did this solution improve the engraftment of mouse satellite cells into the diaphragm? When they tested the hyaluronic acid-gelatin solution in the context of hiPS cell-derived stem cells, the engraftment goes from 1 fiber to 4 fibers: very poor.

8- The bottom line is that there is virtually no engraftment with human cells (immortalized myoblasts and iPSC-derived muscle stem cells), regardless of hyaluronic acid-gelatin solution (5 donor-derived fibers at most, keeping in mind no dystrophin staining was used here). Therefore, the title of the study is misleading, and the results do not support the interpretation of the manuscript.

Minor points:

1- A brief description of the myoblast cell line and the generation of iPS cell-derived myogenic cells should be provided. The reader should not need to look for the author’s previous publications. Because of this lack of information, the description of flow cytometry is meaningless. This section starts as: “...After day 84 of the MuSC induction, the cells….”. The reader has no idea what the authors are referring to.

2- It is not clear whether these invasive studies were performed under an approved animal protocol. This is not mentioned.

3- Description of engraftment analysis on page 11 only mention human SPECTRIN. Incomplete information.

4- Page 9, line 135: can the authors clarify what do they mean with the following statement? “Injection into the diaphragm was recognized by expansion of the injection area”.

6. PLOS authors have the option to publish the peer review history of their article (what does this mean?). If published, this will include your full peer review and any attached files.

Reviewer #1: No

Reviewer #2: No

---

## [Author Response · Author response to Decision Letter 0]

10 Dec 2021

Reviewer #1

Major Concern:

Q1: If diaphragm movements affect cell engraftment, why is this not impacting on mouse MuSC engraftment as well?

A1: In fact, we believe that diaphragm movement does affect the engraftment of primary MuSCs. A previous study (Didier Montarras et al, Science, 2005) reported the transplantation of 1×104 satellite cells into the TA. This number of cells was the same as the number in our study, but the number of fibers derived from the engrafted cells in the TA was nearly ten times higher (301 fibers) than our observations of the diaphragm (40 fibers). Although these data are not comparable, the findings indicate that movement of the diaphragm affects engraftment.

Q2: Authors use a 33G needle for their injections. This is a very small gauge which could potentially lysate the cell suspension being delivered. As human cells are larger in size than mouse cells, this could potentially explain the difference in engraftment. So the authors should perform a simple experiment of cell viability to assess cell death or cell lysis before and after passing through such a small needle, to make sure cell size and cell concentration/density does not negatively impact on the final outcome.

A2: Following the comment, we carried out additional experiments to validate cell proliferation in vitro in 3 groups: 1) passing through the 33G needle and cultured, 2) passing through the needle with a polymer and cultured, and 3) cultured without passing through the needle.

After passing through the needle, the proliferation of primary satellite cells and iMuSCs was decreased, but that of immortalized cells, Hu5/KD3, was not affected.

Moreover, the polymer solution H2G8 increased the cell proliferation of iMuSCs compared with the polymer-free group. We added these results to Fig. 5 and discuss the effects of passing the cells through the needle and the polymer solution on cell proliferation. The positive effect of the polymer suggests a way to enhance the engraftment efficiency of iMuSCs in the diaphragm.

Q3: Graphs with statistical analysis of transplants need to specify the data points / N values as otherwise is difficult to interpret those numbers.

A3: We added N values to the legends of Fig. 2B, 3A, 3C, and 4D.

Q4: The main finding of the study might be overstated in the title, as in actual fact the absolute differences in dystrophin positive fibers per diaphragm is minimal, as shown in figure 4 d-e. It also unclears how such small numbers might result in such a large statistical significance: please clarify, add N values, and consider amending the title into something more factual/descriptive/cautious (e.g., Transplantation and engraftment dynamics of mouse and human myogenic cells in the diaphragm of a mouse model of Duchenne muscular dystrophy)

A4: We thank the reviewer for the comment and agree the title was overstated. Therefore, we changed the title to “Transplantation of human iPSC-derived muscle stem cells in the diaphragm of Duchenne muscular dystrophy model mice”. In the study, we focused on iMuSC transplantation into the diaphragm rather than a comparison with donor cell engraftment. The transplantation of primary satellite cells was successful, but that of iMuSCs without polymer solution (Medium group) was poor, indicating that the protocol for the iMuSC transplantation into the diaphragm needs optimization. H2G8 modestly enhanced the engraftment. We also found the type of polymer influenced the proportion of no h-SPECTRIN positive fibers: 0.56 (Medium group), 0.69 (A2G8 group) and 0.15 (H2G8 group). Thus, polymers are one step at optimizing iMuSC transplantation into the diaphragm. 

We also added N values to the legend of Fig. 4d as noted in our reply to Comment #3.

 

Reviewer #2

Major Concern:

Q1: Fig. 1a: It is impossible to see the presence of PAX7 positive cells as alluded in Fig. 1b. 

A1: We present the ICC staining data separately in Fig. 1A of the revised manuscript to show PAX7 positive cells clearly.

Q2: There is no evidence that the satellite cells are expressing GFP after sorting. The authors are not using GFP to sort satellite cells, and do not show GFP after 3 days in culture.

A2: We added flow cytometry data as supplemental figure S1 to show that the SM/C-2.6 positive lineage marker negative population (purple dots) was GFP positive. We also added the ICC data of GFP to Fig. 1A.

Q3: For the transplantation involving satellite cells and myoblasts, the authors quantify the number of fibers per GFP area. Does it correspond to each injection point? The GFP signal looks greater in the image showing the whole diaphragm when compared to the actual number of fibers (autofluorescence?).

A3: As the reviewer pointed out, the GFP signal looks stronger in the image with the whole diaphragm. Due to the CAG promoter, the EGFP signal is very strong in the primary satellite cells we used. Therefore, the GFP signal was very high.

Q4: Fig. 1c-d: It is hard to tell where exactly cells were injected (panel c, left). Also, many GFP+ myofibers are dystrophin negative. How do the authors explain this discrepancy? How about Pax7 staining? Not clear what we should be looking at for this. In addition, this experiment is missing proper controls.

A4: We apologize for the incomplete data in our figures. We note the injection area with arrowheads in Fig. 1C. We also provide IHC data separately as Fig. 1D to identify all GFP positive fibers as dystrophin positive. The PAX7 staining data in original Fig. 1D is not included in the revised manuscript. Additionally, we added IHC data of the vehicle injection side in Fig. 1D as a control.

Q5: Page 14, line 221: The authors refer to unpublished results for the proof-of-concept that used immortalized myoblasts are able to contribute to muscle regeneration in vivo, but this information seems critical for the present manuscript since the authors barely see engraftment in the diaphragm (5 fibers is negligible). The authors barely describe these results.

A5: We apologize for our poor explanation. An older report found that Hu5/KD3 are able to contribute to muscle regeneration in vivo (Shiomi et al. Gene Therapy, 2011, cited as ref#21). We cited this paper in the sentence instead of our unpublished data.

Q6: Figure 2: Same issue as figure 1. Most fibers do not express DYSTROPHIN. The number of mice per experimental group is missing.

A6: Thank you for the comment. We measured h-SPECTRIN positive fibers in serial sections of the sample in Fig. 2 and confirmed that many GFP positive myofibers are h-SPECTRIN negative (Fig. R1). Thus, we confirmed that the discrepancy of GFP expression and DYSTROPHIN expression were not due to the insufficient immunostaining. One possibility for this observation is the different expression levels of GFP and DYSTROPHIN. The GFP expression was promoted by the strong EF1a promotor in the piggyBac vector, which was multiply transduced, while DYSTROPHIN expression was promoted using an endogenous DYSTROPHIN promotor. Moreover, an interval of 4 weeks after the transplantation might not be enough to see DYSTROPHIN or h-SPECTRIN positive fibers in the Hu5/KD3 transplantation.

Finally, we added the number of mice per experimental group in the appropriate legends.

Fig. R1. 

A representative IHC image of the diaphragm 4 weeks after the transplantation of GFP-transgenic Hu5/KD3. The section was stained with anti-hSPECTRIN antibody (red) and anti-GFP antibody (green). Nuclei were stained with DAPI (blue). Scale bars, 50 μm.

Q7: Experiments with hyaluronic acid-gelatin solution are extremely weak. The authors inject immortalized myoblast into the TA to identify this combination, then, when translating into the diaphragm, they use myoblast first and just quantify the GFP signal, they don t show the number of fibers. Why testing human immortalized myoblasts into the TA instead of the diaphragm? There is no evidence that the TA needs extra-help since it is not a moving muscle. This brings another point, did this solution improve the engraftment of mouse satellite cells into the diaphragm? When they tested the hyaluronic acid-gelatin solution in the context of hiPS cell-derived stem cells, the engraftment goes from 1 fiber to 4 fibers: very poor.

A7: Our aim of this project was to assess the engraftment potential of iMuSCs in the diaphragm of DMD model mice. Therefore, we focused on data regarding iMuSC transplantation into the diaphragm in the revised manuscript and changed the title to: Transplantation of human iPSC derived-muscle stem cells in the diaphragm of a Duchenne muscular dystrophy model mice. 

Since the transplantation of primary satellite cells was successful, we concluded that the poor result for the iMuSC transplantation without polymer solution (Medium group) indicates the need to optimize the protocol. H2G8 modestly improved the engraftment, as the proportion of no h-SPECTRIN positive fibers was smaller (0.15 vs. 0.56 for the Medium group and 0.69 for the A2G8 group). However, H2G8 is just one step in the optimization of iMuSC transplantation into the diaphragm. 

The use of the TA to assess the polymer solution has some advantages. Since the myofibers of the TA are in one direction, it is easy to quantify their regeneration. We then applied our TA findings to iMuSC transplantation to the diaphragm.

As the reviewer mentioned, it would be interesting to see if H2G8 also enhances the engraftment of primary mouse satellite cells. However, in the revised manuscript, we focused on the efficiency of iMuSC transplantation into the diaphragm rather than on the effect of H2G8.

Q8: The bottom line is that there is virtually no engraftment with human cells (immortalized myoblasts and iPSC-derived muscle stem cells), regardless of hyaluronic acid-gelatin solution (5 donor-derived fibers at most, keeping in mind no dystrophin staining was used here). Therefore, the title of the study is misleading, and the results do not support the interpretation of the manuscript.

A8: We agree that our title was inappropriate. We therefore changed it to: Transplantation of human iPSC derived-muscle stem cells in the diaphragm of a Duchenne muscular dystrophy model mice.

To our knowledge, cell transplantation into the diaphragm of mice has never been reported, and we found iMuSC transplantation especially difficult but could be modestly improved if using a polymer solution.

In the study, we did not use DYSTROPHIN as the marker for the engraftment of iMuSCs, because mdx mice have some revertant fibers that express DYSTROPHIN by natural exon skipping. Such endogenous dystrophin positive fibers make it difficult to assess the engraftment. Therefore, we used human-specific SPECTRIN instead of DSYTROPHIN.

#2 reviewer

Minor concern

Q1: A brief description of the myoblast cell line and the generation of iPS cell-derived myogenic cells should be provided. The reader should not need to look for the author’s previous publications. Because of this lack of information, the description of flow cytometry is meaningless. This section starts as: “...After day 84 of the MuSC induction, the cells….”. The reader has no idea what the authors are referring to.

A1: A brief description was added in the Materials and Methods section. (Page 7, line 108-125)

Q2: It is not clear whether these invasive studies were performed under an approved animal protocol. This is not mentioned.

A2: We added information about the study approval.

(Page 5, line 72-75)

Q3: Description of engraftment analysis on page 11 only mention human SPECTRIN. Incomplete information.

A3: We added the following statement to the description: ‘Myofibers derived from engrafted cells were recognized as h-SPECTRIN positive fibers.’ (Page13, line 207-208). 

Q4: Page 9, line 135: can the authors clarify what do they mean with the following statement? “Injection into the diaphragm was recognized by expansion of the injection area”.

A4: We modified the sentence to read: ‘Injection into the diaphragm was recognized by swelling of the injection site.’ (Page10, line158)

---

## [Decision Letter · Decision Letter 1]

1 Feb 2022

PONE-D-21-27260R1Transplantation of human iPSC-derived muscle stem cells in the diaphragm of Duchenne muscular dystrophy model mice

PLOS ONE

Dear Dr. Sakurai,

Thank you for submitting your manuscript to PLOS ONE. After careful consideration, we feel that it has merit but does not fully meet PLOS ONE’s publication criteria as it currently stands. Therefore, we invite you to submit a revised version of the manuscript that addresses the points raised during the review process.

I am returning your manuscript with 2 reviews. After reading the reviews and looking at the manuscript, I would like you to prepare a revised manuscript in response to the reviewer comments. I expect it to discuss:

1. the discrepancy of GFP expression and DYSTROPHIN expression observed in figure 2 (including the nuclear domain of Dystrophin);

2. The use of human-specific SPECTRIN Abs instead of DSYTROPHIN Abs;

3. To amend the title of figure 4 and the general discussion / conclusion of the article to better reflect the results obtained (going from 1 to 4 spectrin-positive fibers remains very low).

We look forward to receiving your revised manuscript.

Kind regards,

Julie Dumonceaux

Academic Editor

PLOS ONE

Journal Requirements:

Reviewers' comments:

Reviewer's Responses to Questions

**Comments to the Author**

1. If the authors have adequately addressed your comments raised in a previous round of review and you feel that this manuscript is now acceptable for publication, you may indicate that here to bypass the “Comments to the Author” section, enter your conflict of interest statement in the “Confidential to Editor” section, and submit your "Accept" recommendation.

Reviewer #1: All comments have been addressed

Reviewer #2: (No Response)

2. Is the manuscript technically sound, and do the data support the conclusions?

Reviewer #1: Yes

Reviewer #2: No

3. Has the statistical analysis been performed appropriately and rigorously? 

Reviewer #1: Yes

Reviewer #2: Yes

4. Have the authors made all data underlying the findings in their manuscript fully available?

Reviewer #1: Yes

Reviewer #2: Yes

5. Is the manuscript presented in an intelligible fashion and written in standard English?

Reviewer #1: Yes

Reviewer #2: Yes

6. Review Comments to the Author

Reviewer #1: I am satisfied with the revised version of this manuscript , which has improved in quality and should now be of sufficient quality to be published

Reviewer #2: Unfortunately, the revised manuscript still falls short in convincing the reader that meaningful (real) engraftment has been reached with human cells. How can one refer to regenerative potential when transplantation of Hu5/KD3 cells only shows one green fiber (which is not even co-localized with dystrophin)? The authors did not address my concern regarding iMusc transplantation. The explanation provided for not using dystrophin staining is not convincing. Double staining with dystrophin and spectrin antibodies is very doable. Even if there are some revertant fibers in this mouse model, it is possible to identify donor-derived myobers, if they are existent (thus suggesting, lack of expression of DYS). This reviewer also questions the relevance of finding “4” spectrin positive fibers in dystrophin-deficient mice. When the frequency is so low, it is very possible that similar levels will also be found in the negative control, denoting non-specific binding. This is why it is so important to use multiple antibodies.

In summary, the results involving human engraftment are meaningless because engraftment is very poor (at comparable levels of background staining), and on top of that, the authors are not able to show the rescue of the proper marker (dystrophin) in the recipient DMD mouse model used.

Minor: line 258 "mouse" should be added prior to “primary cells”.

7. PLOS authors have the option to publish the peer review history of their article (what does this mean?). If published, this will include your full peer review and any attached files.

Reviewer #1: No

Reviewer #2: No

---

## [Author Response · Author response to Decision Letter 1]

14 Mar 2022

In the revision, we have amended the title of figure 4 and discussion to reflect our results obtained and added immunostaining data to explain why we used the human-SPECTRIN signal instead of the DYSTROPHIN signal for our analysis. In separate file, we provide our detailed replies to all the reviewers’ comments.

---

## [Editor Report · Decision Letter 2]

21 Mar 2022

Transplantation of human iPSC-derived muscle stem cells in the diaphragm of Duchenne muscular dystrophy model mice

PONE-D-21-27260R2

Dear Dr.  Sakurai,

We’re pleased to inform you that your manuscript has been judged scientifically suitable for publication and will be formally accepted for publication once it meets all outstanding technical requirements.

Kind regards,

Prof Julie Dumonceaux

Academic Editor

PLOS ONE

---

## [Editor Report · Acceptance letter]

25 Mar 2022

PONE-D-21-27260R2 

Transplantation of human iPSC-derived muscle stem cells in the diaphragm of Duchenne muscular dystrophy model mice 

Dear Dr. Sakurai:

I'm pleased to inform you that your manuscript has been deemed suitable for publication in PLOS ONE. Congratulations! Your manuscript is now with our production department. 

Kind regards, 

on behalf of

Dr. Julie Dumonceaux 

Academic Editor

PLOS ONE